# Wolfberry genomes and the evolution of *Lycium* (Solanaceae)

You-Long Cao [1,2,22 ✉], Yan-long Li[1,2,22], Yun-Fang Fan[1,2,22], Zhen Li[3,4,22], Kouki Yoshida[5,22], Jie-Yu Wang [6,7], Xiao-Kai Ma[6], Ning Wang[8], Nobutaka Mitsuda [9], Toshihisa Kotake [10], Takeshi Ishimizu [11], Kun-Chan Tsai[12], Shan-Ce Niu[13], Diyang Zhang [6,14], Wei-Hong Sun[6,14], Qing Luo[1,2], Jian-Hua Zhao[1,2], Yue Yin[1,2], Bo Zhang[1,2], Jun-Yi Wang[1,2], Ken Qin[1,2], Wei An[1,2], Jun He[1,2], Guo-Li Dai[1,2], Ya-Jun Wang[1,2], Zhi-Gang Shi[1,2], En-Ning Jiao[1,2], Peng-Ju Wu[1,2], Xuedie Liu[6,14], Bin Liu[6,14], Xing-Yu Liao[6,14], Yu-Ting Jiang[6,14], Xia Yu[6,14], Yang Hao[6,14], Xin-Yu Xu[6,14], Shuang-Quan Zou[6,14], Ming-He Li[6], Yu-Yun Hsiao [15], Yu-Fu Lin[16], Chieh-Kai Liang [17], You-Yi Chen[17], Wan-Lin Wu[16], Hsiang-Chai Lu [6], Si-Ren Lan[6,14], Zhi-Wen Wang[18], Xiang Zhao[18], Wen-Ying Zhong[18], Chuan-Ming Yeh [9,10,19 ✉], Wen-Chieh Tsai [15,16,17 ✉], Yves Van de Peer [3,4,20,21 ✉] & Zhong-Jian Liu [1,2,6,14 ✉]

Wolfberry *Lycium*, an economically important genus of the Solanaceae family, contains approximately 80 species and shows a fragmented distribution pattern among the Northern and Southern Hemispheres. Although several herbaceous species of Solanaceae have been subjected to genome sequencing, thus far, no genome sequences of woody representatives have been available. Here, we sequenced the genomes of 13 perennial woody species of *Lycium*, with a focus on *Lycium barbarum*. Integration with other genomes provides clear evidence supporting a whole-genome triplication (WGT) event shared by all hitherto sequenced solanaceous plants, which occurred shortly after the divergence of Solanaceae and Convolvulaceae. We identified new gene families and gene family expansions and contractions that first appeared in Solanaceae. Based on the identification of self-incompatibility related-gene families, we inferred that hybridization hotspots are enriched for genes that might be functioning in gametophytic self-incompatibility pathways in wolfberry. Extremely low expression of *LOCULE NUBER* (*LC*) and *COLORLESS NON-RIPENING* (*CNR*) orthologous genes during *Lycium* fruit development and ripening processes suggests functional diversification of these two genes between *Lycium* and tomato. The existence of additional *flowering locus C-like* MADS-box genes might correlate with the perennial flowering cycle of *Lycium*. Differential gene expression involved in the lignin biosynthetic pathway between *Lycium* and tomato likely illustrates woody and herbaceous differentiation. We also provide evidence that *Lycium* migrated from Africa into Asia, and subsequently from Asia into North America. Our results provide functional insights into Solanaceae origins, evolution and diversification.

A full list of author affiliations appears at the end of the paper.

The potato family, Solanaceae, consisting of ~95 genera and 2300 species, is widely distributed in tropical and temperate regions of the world and displays unique morphological characteristics and extraordinary ecology[1]. Many species in the family are essential economic crops, and the genomes of several species have been sequenced thus far, including tobacco (*Nicotiana tabacum*)[2], potato (*Solanum tuberosum*)[3,4], tomatoes (*Solanum lycopersicum*, *Solanum pennellii*, and *Solanum pimpinellifolium*)[5–7], peppers (*Capsicum annuum*, *Capsicum baccatum*, and *Capsicum chinense*)[8,9], eggplant (*Solanum melongena*)[10], and two *Petunia* species, *Petunia inflata* and *Petunia axillaris*[11]. However, all these genomes are of herbaceous species, while no genome sequence of a woody Solanaceae has been determined yet.

As a woody genus, Goji (枸杞) *Lycium* of the Solanaceae family, contains ~80 species and these species represent important medicinal and food plants. *Lycium* is distributed in the subtropical regions of Asia, Africa, America, and Australia, presenting a fragmented distribution pattern among the Northern and Southern Hemispheres, including the New and Old Worlds[1,12]. Its disjunct distribution has long been a confusing issue. Two hypotheses have been proposed to explain its segmental pattern, namely, its origin before the breakup of Gondwanaland[13], and its long-distance dispersal[14]. Species within *Lycium* have evolved woody tissue and smaller fleshy fruits compared to those of herbaceous plants and other (berry-bearing) sequenced Solanaceae. Therefore, *Lycium* is a key taxon, and its genome should provide novel insights into key innovations that contribute to our understanding of diversification within the Solanaceae family. Here, we sequenced a total of 13 genomes of perennial woody *Lycium* species using different strategies. PacBio technology was used to obtain a reference-quality genome of *Lycium barbarum* with chromosome-level assembly. PacBio and Illumina sequencing were used to obtain a draft assembly of *Lycium ruthenicum*. We also obtained low-coverage genomes of 11 other *Lycium* species, including six from China, four from western North America, and one from the Middle East (Supplementary Table 1). Comparison of *Lycium* genomes provides insight into how different *Lycium* species have evolved drought tolerance in subtropical areas with high salinity and ultraviolet light irradiation, as well as tasty fruit. Moreover, such comparisons provide further insight into *Lycium* and Solanaceae origins, their (genome) evolution, and species diversification.

## Results and discussion

**Genome sequencing characteristics**. Cytogenetic analysis showed that *L. barbarum* with red fruits contains 24 chromosomes (2n = 2x = 24; Supplementary Fig. 1). Survey analysis indicated that the genome of *L. barbarum* is 1.8 Gb in size and has a high level of heterozygosity of ~1%. To overcome the issue of high heterozygosity for whole-genome sequencing, we developed a haploid *L. barbarum* plant (12 chromosomes) by in vitro pollen culture (Supplementary Note 1). The haploid genome was sequenced by PacBio Sequel sequencing. The final genome assembly is 1.67 Gb, with a contig N50 value of 10.75 Mb (Supplementary Table 2). The quality of the assembly was evaluated using the Benchmarking Universal Single-Copy Orthologs (BUSCO)[15]. The results showed that the completeness of *L. barbarum* genome is 97.75%, indicating that the assembly of the *L. barbarum* haploid genome is relatively complete and of high quality (Supplementary Table 3). Finally, high-throughput/resolution chromosome conformation capture (Hi-C) technology was adopted to assess the chromosome-level assembly of the haploid genome (Supplementary Fig. 2). The lengths of the 12 chromosomes range from 106.53 to 172.84 Mb (Supplementary Tables 4

and 5). For comparing the evolution of traits and drought resistance, the heterozygous diploid genome of *L. ruthenicum* with black fruits was sequenced by Illumina and PacBio technologies, and the assembled sequence has a scaffold N50 value of 155.39 Kb and a contig N50 value of 16.14 Kb (Supplementary Table 2). BUSCO assessment indicated that the completeness of the *L. ruthenicum* genome assembly is 96.80% (Supplementary Table 3). For analysis of the disjunct distribution pattern, we further sequenced 11 other genomes of *Lycium* species with 30-fold coverage (Supplementary Table 1).

We confidently annotated 33,581 and 32,711 protein-coding genes in *L. barbarum* and *L. ruthenicum*, respectively (Supplementary Table 6). BUSCO assessment indicated that the completeness of the *L. barbarum* and *L. ruthenicum* genome annotations is 93.16% and 89.38%, respectively (Supplementary Table 7), suggesting that the annotation of these two genomes can be used as reference genome for the *Lycium* genus. In addition, we identified 151 mRNAs, 1255 tRNAs, 2361 rRNAs, and 1243 snRNAs in the *L. barbarum* genome and 165 mRNAs, 1602 tRNAs, 1728 rRNAs, and 1736 snRNAs in the *L. ruthenicum* genome (Supplementary Table 8).

### Genome evolution and the evolution of Solanaceae
*Evolution of gene families*. We constructed a high-confidence phylogenetic tree and estimated the divergence times of 19 species based on sequences extracted from a total of 259 single-copy gene families (see "Methods" and Supplementary Note 2). We found that *L. barbarum* was sister to *L. ruthenicum* (Fig. 1 and Supplementary Data 18). The estimated divergence time of Solanaceae and coffee (Rubiaceae) was 82.8 Mya. We also determined the expansion and contraction of orthologous gene families using CAFÉ3.1 (ref. [16]). Analyses of gene family expansion and contraction found 887 gene families under expansion in Solanaceae, whereas 395 families became smaller (Fig. 1 and Supplementary Data 1). Among the Solanaceae, 275 gene families were expanded in *Lycium* (12 significantly), compared to 715 (24 significantly) that were expanded in *Petunia*, 9645 (487 significantly) that were expanded in *Nicotiana*, 679 (40 significantly) that were expanded in *Capsicum*, and 32 (3 significantly) that were expanded in *Solanum*. Simultaneously, 672 (9 significantly) gene families became smaller in *Lycium* compared to 1009 (11 significantly) that became smaller in *Petunia*, 494 (2 significantly) that became smaller in *Nicotiana*, 1202 (15 significantly) that became smaller in *Capsicum*, and 288 (12 significantly) that became smaller in *Solanum* (Supplementary Data 1).

Significantly expanded gene families of Solanaceae were especially enriched in Gene Ontology (GO) terms like "photosynthesis, light harvesting" and "metabolic process" (Supplementary Data 2), and KEGG terms such as "photosynthesis—antenna proteins" and "metabolic pathways" (Supplementary Data 3). GO enrichment analyses indicated that 232 unique Solanaceae gene families are specifically enriched in GO terms, including "ADP binding", "glycerolipid biosynthetic processes", "response to desiccation", "transcription regulatory region DNA binding", and "dioxygenase activity" (Supplementary Data 4). GO enrichment analyses indicated that 926 unique *L. barbarum* genes are specifically enriched in the GO terms of "deadenylation-independent decapping of nuclear-transcribed mRNA" and "ribonucleoprotein granule" (Supplementary Data 5). For the 433 expanded gene families (34 significantly) in *L. barbarum*, we found enrichment for the GO terms "transferase activity, transferring phosphorus-containing groups", "kinase activity", and "catalytic activity" (Supplementary Fig. 3 and Supplementary Data 6). Enriched KEGG terms were "photosynthesis", "RNA

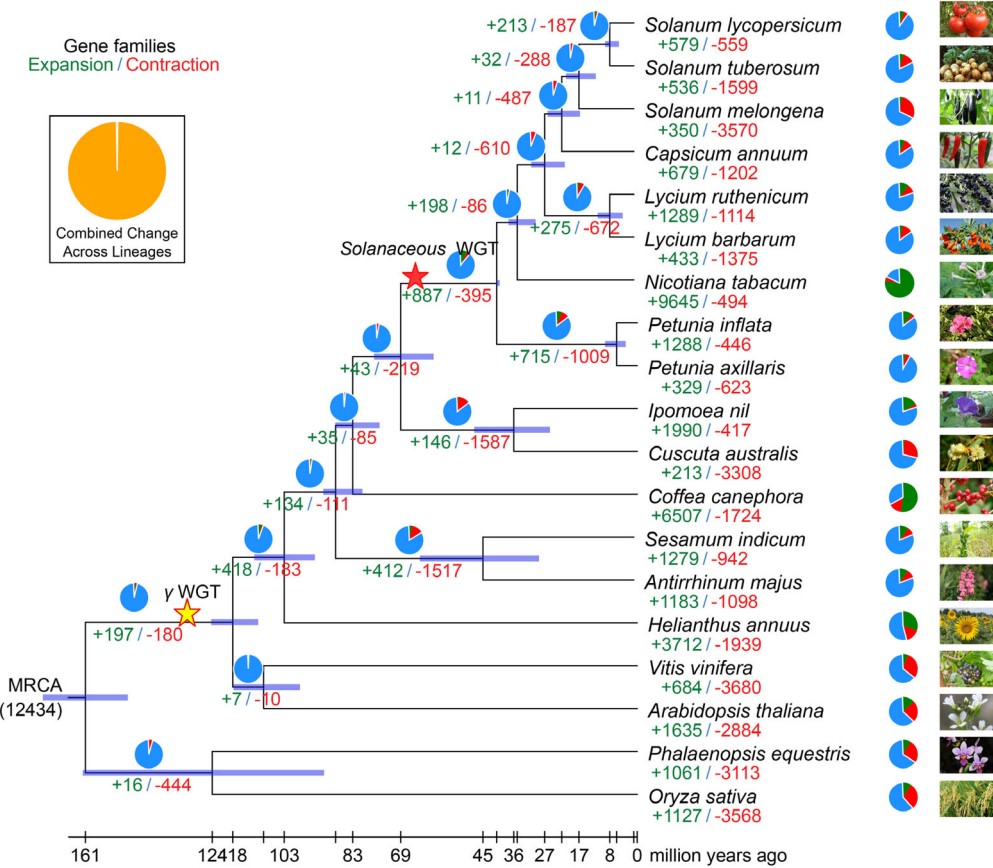

**Fig. 1 Phylogenetic tree showing divergence times and the evolution of gene family size.** The phylogenetic tree shows the topology and divergence times for 19 plant species. As expected, as a genus of the family Solanaceae, *Petunia* is sister to other genera; *Lycium* is sister to *Capsicum* and *Solanum*, indicating that woody *Lycium* evolved from an herb. In general, the estimated Solanaceae divergence times are in agreement with recent broad-scale Solanaceae phylogenies[5]. Divergence times are represented by light blue bars at internodes; the range of these bars indicates the 95% confidence interval of the divergence time. Numbers at branches represent the expansion and contraction of gene families (see "Methods"). MRCA most recent common ancestor. The number in parentheses is the number of gene families in the MRCA as estimated by CAFÉ[16]. The yellow pentagram means the gamma WGT shared by eudicot plants, which occurred ~130 Mya[67]. The red pentagram means the Solanaceous shared WGT, which was estimated at an average of 69 Mya (Supplementary Table 12).

polymerase", "anthocyanin biosynthesis", and "phenylpropanoid biosynthesis" (Supplementary Fig. 4 and Supplementary Data 7).

*Lycium* species are absent in tropical regions, resulting in a fragmented distribution at high altitudes in the subtropics to temperate regions. We compared *Lycium* with other Solanaceae species, to identify genes that might be associated with adaptation to strong light, arid, and saline environments (Supplementary Fig. 5). *Lycium* displayed significant expansion of the triose phosphate/phosphate translocator (TPT) gene family. During photosynthesis, triose is exported from the chloroplasts to the cytosol and is used for sucrose synthesis. TPT is a plastidic transporter and it exchanges triose phosphate with phosphate. *Arabidopsis*' loss of function TPT mutants had starch accumulating in their chloroplasts, and both photosynthesis and growth decreased[17]. This process is essential for optimal photophosphorylation in the chloroplast. The export of photosynthetic products from the chloroplast via TPT is crucial for the maintenance of high rates of photosynthetic electron transport. The high copy number and specific loss of the N-terminal domain for *Lycium* species would relate to their resistance to intense light.

Abscisic acid also plays an important role in response to environmental stresses especially in dry and highly saline conditions[18]. We identified genes involved in the biosynthetic pathways for Solanaceae species, including two *Lycium* species

(Supplementary Fig. 6). Zeaxanthin epoxidase (ZEP) is the first enzyme for the ABA biosynthetic pathway; it converts zeaxanthin into antheraxanthin and subsequently violaxanthin. It had a specific expansion in *Lycium* and resulted in an extra subclade that shows a sister relationship with other Solanaceae homologs with special motifs (Supplementary Fig. 6a). ABA2 is another gene that participates in ABA biosynthesis. It encodes xanthoxin oxidase, which is converted into xanthoxin, the precursor of ABA and ABA-aldehyde. *L. ruthenicum* and *L. barbarum* displayed divergence in ABA2 copy number and sequence structure, indicating dissimilar ABA biosynthetic pathways.

**Hexaploidization in Solanaceae.** Distributions of synonymous substitutions per synonymous site ($K_S$) for the whole paranome (all duplicate genes) and the anchor pairs (duplicate genes retained in collinear blocks) in both *L. barbarum* and *L. ruthenium* show clear peaks at $K_S \approx 0.65$ (Supplementary Fig. 7), suggestive of an ancient polyploidization event in the two *Lycium* genomes. In addition, peaks with similar $K_S$ values could be identified in the paranome $K_S$ distributions in other Solanaceae species, such as *S. lycopersicum*, *N. tabacum*, and *P. axillaris*, and in Convolvulaceae species, such as *Ipomoea nil* (Supplementary Fig. 8). Intergenomic comparisons between the genomes of *L. barbarum* and *Vitis vinifera* show patterns of collinearity consistent with two hexaploidization events, a more recent one and an older one

(Supplementary Fig. 9). Similarly, remnants of hexaploidization events have also been observed for *S. lycopersicum*[5] and *Ipomoea* genomes[19]. To test whether the most recent hexaploidization could have been shared between Solanaceae and Convolvulaceae, we used one-to-one orthologous $K_S$ distributions to compare synonymous substitution rates of Solanaceae and Convolvulaceae species, as well as information on the speciation events between *L. barbarum* and *S. lycopersicum*, *N. tabacum*, *P. axillaris*, *I. nil*, and *Coffea canephora*. Similar $K_S$ distances of the investigated Solanaceae and Convolvulaceae species to *C. canephora* indicate that they have similar synonymous substitutions (Supplementary Fig. 10); therefore, the orthologous $K_S$ distributions between *L. barbarum* and the studied species can be used to compare with the paralogous $K_S$ distribution in *L. barbarum* without having to correct for different synonymous substitution rates. The peak in the $K_S$ distributions of anchor pairs in *L. barbarum* is younger than the $K_S$ peak representing the divergence between *Lycium* and *Ipomoea*, but older than the $K_S$ peak representing the speciation events between *Lycium* and other Solanaceae species (Fig. 2 and Supplementary Data 19). Therefore, our data confirm that the *Lycium* genomes and other sequenced Solanaceae species share the same hexaploidization event, which occurred after the divergence of Solanaceae and Convolvulaceae[19].

**Self-incompatibility in wolfberry.** Self-fertilization is prevented by S-RNase-mediated gametophytic self-incompatibility (GSI) in some species of the Solanaceae family[7,20]. S-RNases belong to the T2-type RNases (RNases-T2) that are responsible for self-pollen recognition and rejection in Solanaceae[21]. The homologous genes of S-RNase in wolfberry were identified. To search for genes potentially involved in wolfberry GSI, through homology searches and phylogenetic reconstruction[22], RNase-T2s genes were divided into three subfamilies, namely, class I RNase-T2s, class II RNase-T2s, and S-RNases. The S-RNases can be further subdivided into four groups as Rosaceae S-RNases, Antirrhinum S-RNases, Solanaceae S-RNases I, and Solanaceae S-RNases II (Supplementary Fig. 11). Among all 17 RNase-T2s genes identified in *L. barbarum*, 2 and 15 genes are clustered as class I RNase-T2s and S-RNases, respectively. The latter includes 14 genes in Solanaceae S-RNases II and one gene in Solanaceae S-RNases I (*Lba02g01102*), and two genes class I (Supplementary Fig. 11 and Supplementary Data 8). The gene *Lba02g01102* also clustered

together with S-RNase genes from *Lycium* species studied by Miller et al.[23] (Supplementary Fig. 12), which further confirmed its S-RNase activity. Previous evidence showed that canonical Solanaceae S-RNases I is responsible for the self-incompatibility within the Solanaceae S-RNases groups; however, Solanaceae S-RNases II may be part of the *Petunia* and tobacco nectar defense repertoire[24]. Therefore, the gene family analysis showed that the gene *Lba02g01102* might be the most likely candidate controlling the GSI in wolfberry. In addition, using key words (F-box) and HMMER search (PF00646.34) methods, 49 F-box genes were found on chromosome 2. Among them, ten were annotated as S-locus-linked F-box genes (blue color) located in the flanking region of gene *Lba02g01102* (green color, Supplementary Fig. 13), suggesting this is the S-locus candidate.

SI genes usually contribute to and maintain the genetic variation and high level of heterozygosity in SI species, therefore maintaining heterosis[11,25–27]. The diploid *L. barbarum* genome has a high level of heterozygosity (1.0%) that might be contributed to or maintained by SI genes. To further investigate their relationship and whether the RNase-T2s genes are mainly located in hybridization hotspots and contribute to the high level of heterozygosity of the wolfberry genome, a sliding windows approach was used to find hybridization hotspots along the 12 chromosomes of the *L. barbarum* genome (Supplementary Fig. 14). The heterozygous sites were identified using the genotype calling of a diploid individual using 20× Illumina reads mapping to the haploid reference genome. The criteria for detecting the hybridization hotspots included a 200-kb sliding window with >1000 heterozygous sites, representing a >0.5% heterozygosity rate in each window (see "Methods" and Supplementary Fig. 15). The identified 2782 hybridization hotspots are not distributed randomly over the genome and account for 60.03% of the heterozygous sites for the entire genome. The distribution of heterozygous sites in chromosome 2 has different patterns from other chromosomes, showing elevated heterozygous rates in the middle region of the chromosome (the highlighted 40–100 Mb region along chromosome 2 in Supplementary Fig. 16). Chromosome 2 also shows the highest number of heterozygous sites per hotspot (Supplementary Fig. 17), indicating the highest recombination frequency in chromosome 2 than others.

Genes in these 2782 hotspot regions are mainly enriched in GOs with amino acid and sulfur compound metabolic processes (biological processes; $P < 0.05$), vacuole and membrane-bound organelles (cellular components; $P < 0.05$), and oxidoreductase activity (molecular function; $P < 0.05$; Supplementary Fig. 18). Among these genes, nine of them, present in the six hybridization hotspots along chromosomes 2, 5, 6, 7, and 11, mainly function in RNase-T2 activity (Supplementary Fig. 14 and Supplementary Data 9).

The above results indicate that the S-RNases I type SI candidate (*Lba02g01102*) gene might be the most likely candidate key gene controlling the GSI in wolfberry. In addition, the frequent recombination on chromosome 2 might be maintained by this SI gene, with the direct effect of an elevated heterozygosity rate in SI wolfberry. However, the molecular mechanism of recombination elevation caused by this candidate gene requires further study.

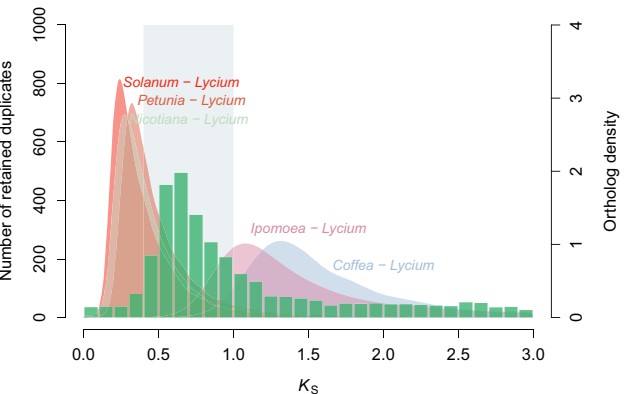

**Fig. 2 Whole-genome duplication in Solanaceae.** $K_S$ age distributions for anchor pairs of *L. barbarum* (green histogram; left hand *y*-axis; a peak represents an ancient polyploid event) and for one-to-one orthologues between *L. barbarum* and *S. lycopersicum*, *N. tabacum*, *P. axillaris*, *I. nil*, and an outgroup species *C. canephora* (colored filled curves of kernel-density estimates; a peak represents a species divergence event). The gray rectangles highlight the peak found in the anchor pair $K_S$ distributions with a range of 0.4–1.0.

**Genes involved in secondary cell wall growth.** *L. barbarum* is a perennial deciduous shrub species with woody stems[28,29]. Thus far, all species that have been sequenced in Solanaceae are herbaceous plants, such as tomato, tobacco, potato, pepper, eggplant, and *Petunia*. We were therefore interested to analyze genes involved in secondary cell wall (SCW) formation because the development of woody stems in higher plants is associated with

the SCW, which is primarily composed of cellulose, hemi-cellulose, and lignin[30,31]. The putative cellulose and lignin bio-synthetic genes in *L. barbarum* are listed in Supplementary Data 10 and 11 (refs. [31,32]). Interestingly, many tandem gene duplications were uncovered, especially in the cinnamyl alcohol dehydrogenase (*CAD*) and laccase (*LAC*) gene families, which are involved in the final two steps of the lignin biosynthesis pathway (Supplementary Data 11). We further examined the expression of these tandem duplicated genes in the four transcriptomes prepared from the basal to apical regions of the stems. As the data show in Supplementary Data 12, most of these genes were relatively highly expressed in the two apical regions of the stems, which may undergo vigorous SCW formation and lignification, suggesting their potential important roles in lignin biosynthesis. *L. barbarum* is a halophyte grown under extreme conditions, especially under high-saline and dry environments, in arid and semiarid areas of China[33,34]. Lignin provides plants rigidity and is an effective mechanical barrier against pathogen attacks. Its water-impermeable property is known to reduce water tran-spiration and to make it possible to maintain normal turgor pressure under drought conditions[35]. Correlation of lignin deposition in the root regions and drought tolerance has been demonstrated for several plant species[36]. Therefore, it is inter-esting to speculate on these duplicated genes being involved in the adaptation of *L. barbarum* to arid environments.

**Perennial flowering in a life cycle and flower development**. MADS-box transcription factors are among the most important regulators of plant floral development and a major class of reg-ulators mediating floral transition. The *L. barbarum* genome encodes 80 MADS-box genes, including 25 type I and 55 type II MADS-box genes (Supplementary Data 13). The number of MADS-box genes is greater than that in *Amborella* and in the primitive orchid *Apostasia*, but lower than in the two *Solanum* plants, tomato (131 MADS-box genes) and potato (153 genes; Table 1). Notably, tomato (81 members) and potato (114 mem-bers) contain 3.24- and 4.56-fold gene numbers in type I MADS-box genes, respectively, compared to *L. barbarum* (25 members; Supplementary Data 13). Tandem gene duplications seem to have contributed to the increase in the number of type I genes in the α group (type I Mα) and suggest that type I MADS-box genes have mainly been duplicated by smaller-scale and more recent duplications[37]. Most type II MADS-box gene subfamilies exist in tomato, potato, and *L. barbarum*, except the *OsMADS32* sub-family (Supplementary Fig. 19), which has been lost at the base of extant eudicots[38]. Interestingly, we found that *L. barbarum* has more members (nine members) in the *FLOWERING LOCUS C* (*FLC*) clade than tomato (three members) and potato (two members; Supplementary Data 13). *FLC* of *Arabidopsis thaliana* is a well-studied and important repressor of floral transition,

which blocks flowering until plants are exposed to the winter cold. It also has been documented that allelic variation at the *FLC* locus contributes to flowering variation in *Arabidopsis lyrata*, *Arabis alpine*, and *Brassica oleraceae*[39]. Resetting of *A. alpina FLC* transcription after vernalization in vegetative tissue is important for determining the perennial life cycle duration of flowering and maintaining vegetative axillary branches that flower the following year[40]. We proposed that the expanded *FLC*-like genes with sequence variation in *L. barbarum* might be related to the per-ennial flowering cycle of *Lycium*. Further studies are necessary on the function of *Lycium FLC* genes for flowering.

The *L. barbarum* genome reveals a comparable number of floral organ identity genes to that of tomato and potato (Supplementary Data 13). We found that expression of A-class genes *Lba01g02228* and *Lba03g02360*, B-AP3 genes *Lba12g01999* and *Lba02g02684*, B-PI genes *Lba01g01562* and *Lba04g01347*, and E-class genes *Lba12g02569* and *Lba05g01643*, is primarily detected in floral developmental stages (Supplementary Fig. 20).

**Fruit development and ripening**. Fruit development and ripening is a complex process that undergoes dramatic changes. These changes mainly involve flavor, color, and texture, which influence fruit quality. Although fruit is the most valuable pro-duct of *Lycium*, very few studies related to the genetic regulation of the fruit development and ripening have been reported. As a member of Solanaceae, *L. barbarum* develops fleshy fruit similar to tomato. The biochemical changes and molecular regulation underlying processes, such as softening, color change, and the regulation of ripening have been depicted based on the study of tomato fruit development[41] and are similar in some respects in *L. barbarum* (Supplementary Fig. 21). The phytohormone ethy-lene regulates and coordinates different aspects of the ripening process[42], and a series of tomato ripening mutants have been isolated and have elucidated the ripening mechanism in fruit plants[42–44]. Not surprisingly, most of the transcription factors involved in fleshy fruit development and ripening could be identified in the genome of *L. barbarum*. These genes' expression profiles fit the processes of fleshy fruit development and ripening, suggesting high conservation of the regulatory program between tomato and *L. barbarum*. Notably, we observed that, at the fruit developmental stage, the C-class gene *Lba02g01441* is pre-dominantly expressed at an early stage of fruit development, and the *Lba11g00615* transcript accumulates mainly at the ripening stage (Supplementary Fig. 20a). In addition, expression of A-class genes *Lba01g02228* and *Lba03g02360*, D-class gene *Lba01g01850*, and E-class genes *Lba12g02569* and *Lba05g01643* was high at fruit developmental stages (Supplementary Fig. 20b). Interestingly, the E-class gene *Lba05g02389* shows differential expression between the floral and fruit developmental stage. and is mainly expressed in fruit developmental stage 2 (Supplementary Fig. 20a). In

**Table 1 MADS-box genes in *A. trichopoda*, *S. lycopersicum*, *S. tuberosum*, *L. barbarum*, poplar, *Arabidopsis*, rice, and orchid *Apostasia* genomes.**

| Category | A. trichopoda[115] | S. lycopersicum[116] | S. tuberosum[117] | L. barbarum[a] | Poplar[118] | Arabidopsis[37] | Rice[119] | Apostasia[120] |
|---|---|---|---|---|---|---|---|---|
| Type II (total) | 23 | 50 | 39 | 55 | 64 | 45 | 44 | 27 |
| MIKC[c] | 21 | 40 | 30 | 50 | 55 | 39 | 39 | 25 |
| MIKC* | 2 | 10 | 9 | 5 | 9 | 6 | 5 | 2 |
| Type I (total) | 13 | 81 | 114 | 25 | 41 | 61 | 31 | 9 |
| Mα | 6 | 62 | 70 | 15 | 23 | 25 | 12 | 5 |
| Mβ | 6 | 6 | 28 | 3 | 12 | 20 | 9 | 0 |
| Mγ | 1 | 13 | 16 | 7 | 6 | 16 | 10 | 4 |
| Total | 36 | 131 | 153 | 80 | 105 | 106 | 75 | 36 |

[a]This study.

addition, the expression levels of *LOCULE NUBER* (*LC*) ortho-logue *Lba12g01956* and *COLORLESS NON-RIPENING* (*CNR*) orthologue *Lba12g01571* (Supplementary Figs. 22–25) were barely detected during fruit development and the ripening process (Supplementary Fig. 20b). *LC* is a homeodomain transcription factor gene controlling the tomato fruit locule number and is flat-shaped[45]. *CNR* is a SBP-box transcription factor gene and is considered to be one of the master regulators of tomato fruit ripening[46]. Extremely low expression of *LC* and *CNR* orthologous genes during *L. barbarum* fruit development and the ripening process suggested that these two genes might be regulated by other processes in *Lycium*, and might be related to the differences in fruit size and shape compared to tomato.

The *NOR* mutant clarified its significant effect on fruit ripening in tomato, in which the gene was identified as a transcription factor encoding a member of the *NAC* transcription factor family, composed of a large number of paralogous genes. Other ripening-related *NAC* coding genes, such as *SlNAC4* (Solyc11g017470.1.1) and *NOR-like1* (Solyc07g064320.2.1), showed high accumulation at the onset of fruit ripening, and repressed gene expression resulted in delayed fruit ripening in tomato. Transcriptome analysis showed a portion of ripening-related *NAC* family genes described above expressed in the examined tissues of *L. barbarum*. We identified *Lba02g00132*, referred to as *NOR*, *Lba02g01723*, referred to as *SlNAC4*, and *Lba05g02503*, referred to as the homolog gene *NOR-like1* in the *L. barbarum* genome. These key ripening factors showed accumulation of transcripts in the root, stem, leaf, and in all stages of fruit development, whereas not *NOR* but *Lba05g02503* (homolog of *NOR-like*, Solyc05g007770.2.1) far exceeded the expression level in *L. barbarum* (Supplementary Fig. 20c), emphasizing the important roles in the functioning of fruit ripening of *Lba05g02503*. Comparisons of gene expression patterns across *L. barbarum* and tomato revealed differences in gene expression and functional capabilities of fruit development and ripening control.

**Genes related to polysaccharide synthesis.** The fruits of *L. barbarum* have been demonstrated to possess multiple biological effects against cancer, inflammation, diabetes, weariness, oxida-tive stress, radiation, and ageing[47,48]. The active components in these fruits mainly include polysaccharides, flavonoids, car-otenoids, zeaxanthins, betaine, and cerebroside[47,49]. Among them, *L. barbarum* polysaccharides (LBP) are the primary active components with a wide array of pharmacological activities. Although many LBPs have been isolated and characterized[29], their detailed structural features have not been fully defined. Recently, two studies reported LBPs as pectin molecules and proposed their hypothetical structures[47,50]. Pectin is one of the three key cell wall polysaccharides, mainly composed of three structural domains, including linear homogalacturonan (HG) regions and branched rhamnogalacturonan type I (RG-I) and RG-II regions[51]. RG-I is often linked by several side chains, such as arabinogalactan (AG) types I and II, while the RG-II structure is more complex[47]. Similar to typical pectin structures in other plants, LBPs have been shown to contain HG, RG, and AG regions[47,50] (Supplementary Fig. 26). *Arabidopsis* genes required for pectin synthesis, including glycosyltransferases (GTs) and methyltransferases (MTs), have been identified or predicted[51–56]. Therefore, we used *Arabidopsis* genes as queries to identify putative *Lycium* genes for the biosynthesis of LBPs (Supple-mentary Fig. 26). Some GT and MT genes required for HG and AG synthesis were also used as queries, including genes encoding QUASIMODOs (*QUAs*), cotton Golgi-related 3 (*CGR3*), hydro-xyproline O-galactosyltransferases (*HPGTs*), fucosyltransferases (*FUTs*), β-glucuronosyltransferases (*GlcAT14A, B, C*), and galactosyltransferases (*GALTs*)[53,54]. The putative genes for LBP biosynthesis in the *Lycium* genome and their expression levels are listed in Supplementary Data 14. Comparison of the numbers of these putative LBP biosynthetic genes in *L. barbarum* with other Solanaceae species was performed (Supplementary Data 15). To identify potential candidate genes, we analyzed the transcriptome and found that the expression patterns of many of the genes during fruit development stages were consistent with the accu-mulation of LBP or peaked prominently at the initial stage, during which LBP content is high, such as *Lba01g02635*, *Lba04g00011*, *Lba12g00767*, *Lba08g01208*, *Lba07g00975*, *Lba03g02350*, and *Lba01g02220* (Fig. 3 and Supplementary Data 14). Interestingly, their *Arabidopsis* orthologues, *GALT29A*, *GALT31A*, *GALS3*, *GAUT7*, and *RRT1*, are key genes encoding enzymes for the biosynthesis of HG, RG-I, and AG[51,53,56]. These findings suggest that these candidate genes may play an impor-tant role in LBP biosynthesis. Moreover, a possible model of the biosynthetic pathways of LBP was proposed based on *Arabidopsis* research (Supplementary Fig. 26a), and putative *Lycium* genes involved in the pathway were identified through their homology with genes in the *Arabidopsis* genome (Supplementary Data 16). Tandem gene duplicates of *SWEET17* (*Lba06g02490* and *Lba06g02491*) were identified in the *Lycium* genome (Supple-mentary Data 16). It is interesting to know if these duplicated genes can contribute to LBP biosynthesis in *Lycium*. In addition, the numbers of *Lycium* genes in this biosynthetic pathway were compared with other Solanaceae genomes (Supplementary Data 17a). There are no significant differences between *Lycium* and other Solanaceae species in most genes listed in Supple-mentary Data 17a. However, the number of the whole family genes of *SWEET* in the *Lycium* genome is higher than all Sola-naceae species analyzed in this study, except that of *Petunia* (Supplementary Data 17a). Interestingly, within the 37 *SWEET* genes in the *Lycium* genome, 23 of them are tandem duplicates from seven sets of duplications (Supplementary Data 17b). Fur-ther studies are required to address their functions and to dis-tinguish whether these gene duplications act as a mechanism of genomic adaptation to extreme environments suffered by *Lycium*.

**Genes involved in the different metabolic phenotypes between the two *Lycium* species' fruits**

*Anthocyanin biosynthetic pathways.* Anthocyanin is a kind of water-soluble pigment that is widely found in plant petals, fruit tissue, stems, and leaves. Due to its potent antioxidant and free radical scavenging ability, anthocyanins are often used as nutri-tious food additives. The fruits of *L. ruthenicum* is one of the fruits with the highest anthocyanin content; its anthocyanin content is ~1.7-fold compared to that of the blueberry fruit[57]. The petunidin-3-O-rutinoside (trans-p-coumaroyl)-5-O-glucoside is the most abundant anthocyanin and accounts for 60% of the total anthocyanins[58]. The expression profiles of genes (Supplementary Tables 9 and 10) involved in the anthocyanin biosynthetic pathways (ABP) were compared during the fruit development and ripening processes of *L. barbarum* and *L. ruthenicum* (Sup-plementary Fig. 27). The results indicated that expression of these genes is significantly higher in *L. ruthenicum* fruits than in *L. barbarum* fruits (Supplementary Fig. 27). In addition, the transcripts of these genes predominantly accumulated at stage 3 to stage 5 of the fruit ripening process (Supplementary Fig. 27 and Supplementary Table 10). The expression patterns of ABP structure genes were correlated with the pigmentation process of the *L. ruthenicum* fruit. Interestingly, there were also obvious differences in the expression levels of four *R2R3-MYB* genes (*Lba04g02548*, *Lba05g02024*, *Lba08g01154*, and *Lba05g02025*) between the two *Lycium* fruits (Supplementary Fig. 27c), which

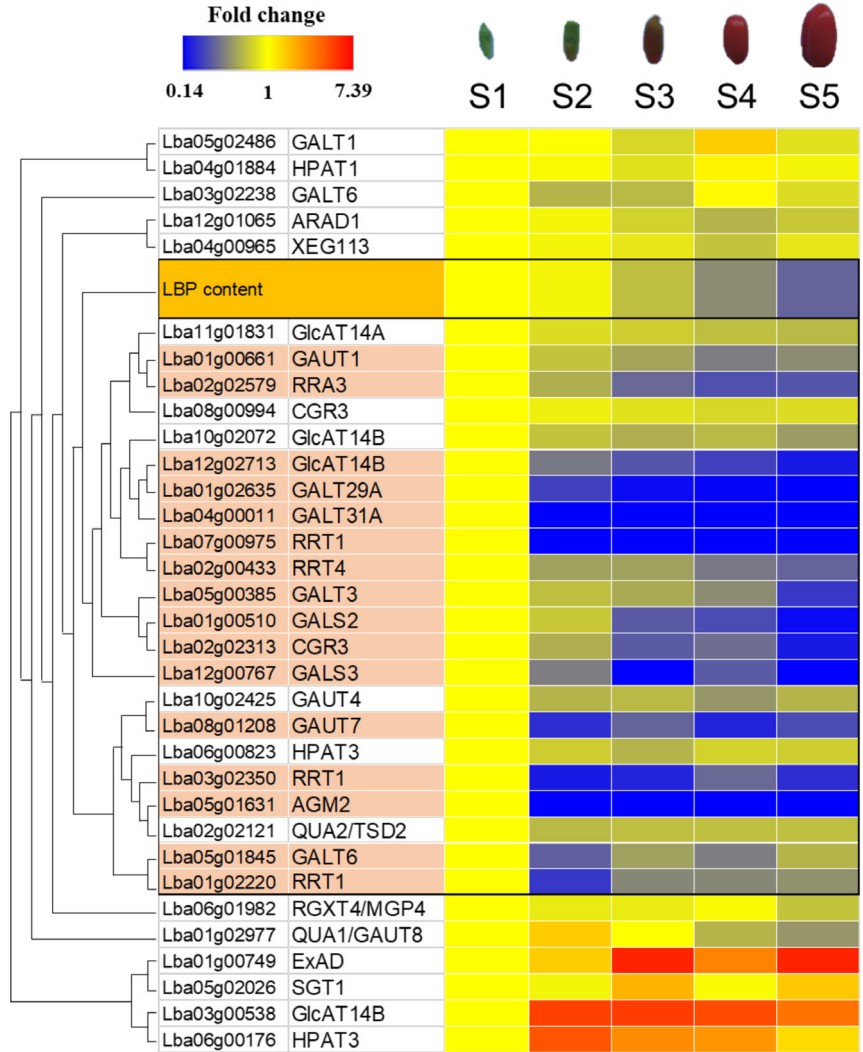

**Fig. 3 Heat map of *L. barbarum* polysaccharide (LBP) and expression of putative genes responsible for LBP biosynthesis during fruit development in *Lycium*.** Fruit development was divided into five stages (S1–S5), as shown at the top of the figure. LBP contents or fragments per kilobase million (FPKM) values of all genes in the S1 stage were set to 1, and the fold changes of LBP contents or FPKM for the other stages (S2–S5) were calculated. LBP contents and gene expression patterns during fruit development are shown in red (high)–yellow–blue (low) gradation. Only FPKM fold changes >0.5 were considered as downregulation. Genes marked in light orange represent candidate genes for LBP biosynthesis. The localization of these gene products and the relationship of GTases with LBP structures are indicated in Supplementary Fig. 26. S1: young stage, 9 days post anthesis; S2: green stage, 15 days post anthesis; S3: turning stage, 21 days post anthesis; S4: red stage, 28 days post anthesis; S5: ripe stage, 35 days post anthesis.

may be involved in the regulation of anthocyanin synthesis of *L. ruthenicum*[48,59].

**Carotenoid biosynthetic pathways.** Genes involved in the carotenoid biosynthetic pathways were identified, and their expression profiles were compared between *L. barbarum* and *L. ruthenicum* fruit. As shown in Supplementary Fig. 28, the expression levels of *ZISO* (*Lba07g02021*), *LCYB* (*Lba05g00383*), *CHYB* (*Lba03g01505*), *PDS* (*Lba03g03127*), *PSY* (*Lba11g02324*), and *ZDS* (*Lba06g01695*) were much higher in *L. barbarum* than those in *L. ruthenicum*, which was in accordance with a previous report[60]. In addition, the expression levels of *CAROTENOID-CLEVAGE DIOXYGENASES* (*CCDs*), which degrade carotenoids, were increased at stage 3 to stage 5 of fruit development of *L. ruthenicum* (Supplementary Fig. 28b). However, the expression of *CCD* (*Lba03g01270*) was only majorly observed at stage 1 in *L. barbarum* fruit (Supplementary Fig. 28b). These results suggest that both regulations of biosynthesis and degradation of

carotenoids are important to the high content of carotenoid accumulation in *L. barbarum* fruit.

Zeaxanthin dipalmitate (ZD) is the main form of carotenoids in *L. barbarum* fruit[61], but the enzymes required for the palmitate esterification of zeaxanthin are not known. Palmitate is a 16-carbon fatty acid. In our study, we found that the expression of a gene (*Lba07g02085*) was gradually increased during the fruit maturation in *L. barbarum*. Our further analysis with the "NCBI CD-Search showed that the gene (*Lba07g02085*) encoded a protein containing a ttLC_FACS_AEE21_like domain. The enzyme family with proteins containing ttLC_FACS_AEE21_like domains can activate" medium and long-chain fatty acids[62,63], and therefore this gene (*Lba07g02085*) was named as *FACS* (*Fatty Acyl-CoA synthetase*). In *A. thaliana*, nine long-chain acyl-CoA synthetase (*ACS*) enzymes were shown to be able to bind 16-carbon fatty acids and unsaturated 18-carbon fatty acids, but not with eicosenoic acid (20: 1) and stearic acid (18: 1). This would suggest that the 16-carbon unsaturated fatty acid palmitate is the best substrate for *ACS*[64]. In the comparative transcriptome

analysis above, we identified that the expression level of *FACS* in *L. barbarum* was almost ten times higher than that in *L. ruthenicum* (Supplementary Table 11). Thus, we inferred that *FACS* (*Lba07g02085*) could play a key role in catalysing the biosynthesis of ZD.

**Biogeography of *Lycium*.** The genus *Lycium* has a fragmented distribution in the subtropical regions of Asia, Africa, North and South America, and Australia and its disjunct distribution has puzzled researchers for a long time. The estimated divergence time of *Lycium* from other Solanaceae species is much later than the breakup of Gondwanaland and the hypothesis that *Lycium* originated before the breakup of Gondwanaland appears incorrect[13,65]. Long-distance dispersal[14] seems to be the best explanation for the current distribution of *Lycium* species. Based on the hypotheses of Fukuda et al.[65] and Miller et al.[66], *Lycium*

originated in south America, then migrated in different directions into North America and Africa, and finally into Eurasia, Australia, and eastern Asia.

We constructed a phylogenetic tree for *Lycium* species that represent both Old World and North America representatives based on 2,992,053 high-quality single-nucleotide polymorphisms (SNPs; Fig. 4a and Supplementary Data 20). In our phylogenetic tree, all Asian species clustered into a single clade, except for *L. depressum* from the Middle East and the widely spread species, *L. ruthenicum* (Fig. 4a), where *L. ruthenicum* is the first diverging species followed by the Middle East species (*L. depressum*). North American species also form a single clade and nest within Old World *Lycium* with strong support. This relationship was also supported by principal component analysis (PCA; Fig. 4b and Supplementary Data 20). We estimated the historic population size of *L. babarum* based on the pairwise sequentially Markovian

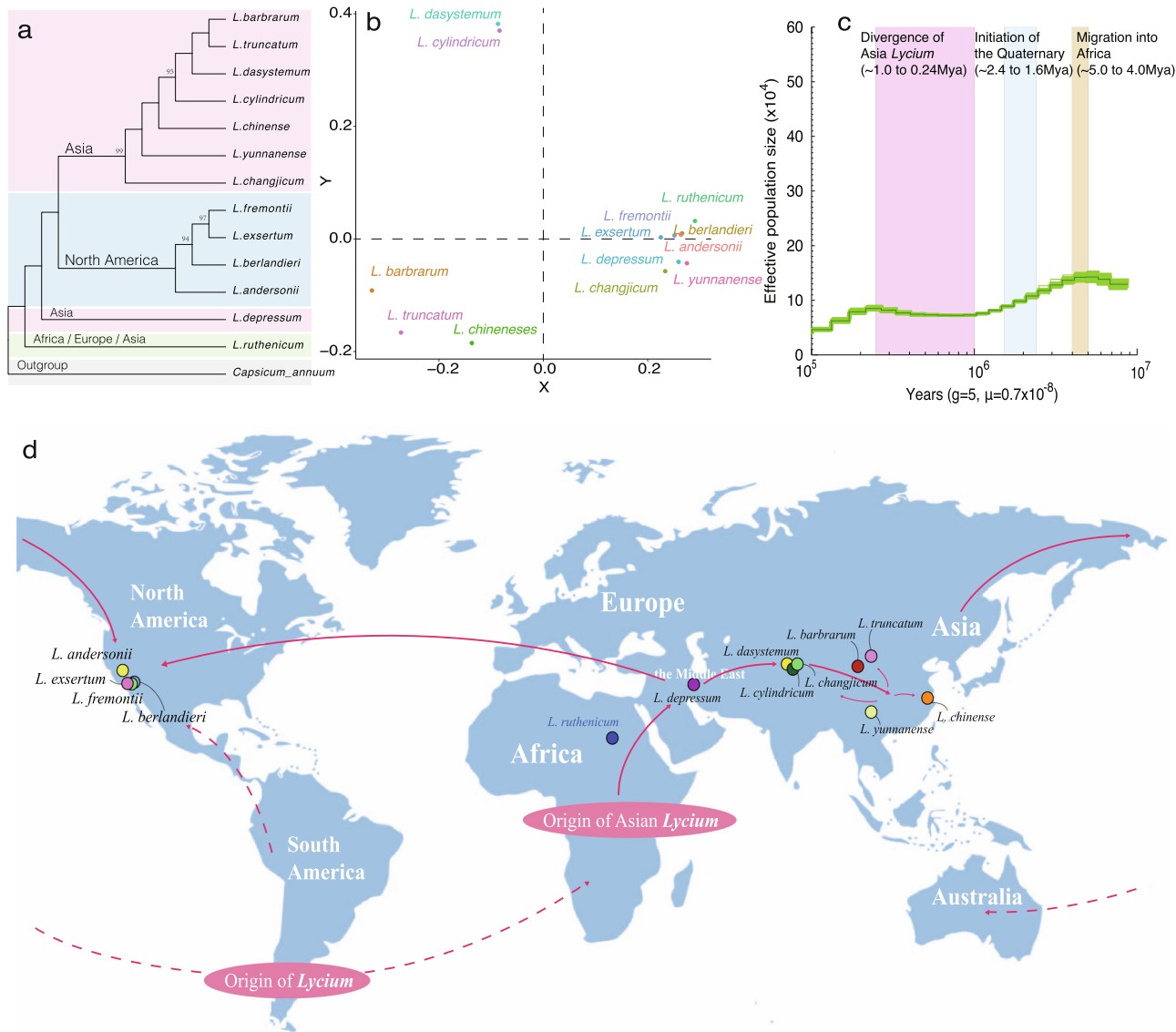

**Fig. 4 Biogeography and evolution of *Lycium*. a** Maximum likelihood phylogenetic tree of *Lycium* species was constructed based on single-nucleotide polymorphisms (SNP) loci. Only bootstraps <1.00 are displayed at the nodes. **b** Principal component analysis (PCA) of *Lycium* species showing the relationships among different species. The blue or red dots represent species sample locations in Asia or North America, respectively. **c** Effective population sizes of *L. barbarum* based on pairwise sequentially Markovian coalescent (PSMC); the *x*-axis indicates the years before recent. The generation and substitution rates were set at 5 and $0.7 \times 10^{-8}$, respectively. **d** Proposed origins and ancient dispersal routes of *Lycium*. Arrows indicate possible migration directions of *Lycium* species. The dashed lines are based on the hypothesis from Miller et al.[66] and Fukada et al.[65] and the full lines indicate the hypothesis from our study.

coalescent method (PSMC; Fig. 4c), and found a *Ne* decreasing event in *Lycium* ~5.0–4.0 Mya, and an increasing event at ~1.0 Mya. These time points correspond to a possible bottleneck during the migration from American to Africa, and the divergence of eastern Asia *Lycium*, as mentioned by Miller et al.[66].

In the phylogenetic tree, *L. ruthenicum* is the first diverging species followed by *L. depressum*. *L. ruthenicum* is a widely distributed species that occurs throughout northern Africa, south-eastern Europe, and north-western China, while *L. depressum* can be found in the Middle East. The African *Lycium* could be derived from the South American *Lycium*, while it could be the ancestor of the Asian *Lycium*. During the migration from Africa to Asia, *L. ruthenicum* most likely first evolved in Africa, after which the speciation of *L. depressum* occurred in the Middle East during migration toward Asia (Fig. 4d).

The north American species and eastern Asian species cluster together and display a sister relationship, while both of them show a close relationship to the Middle East species. In the Asian clade, the *L. yunnanense* and *L. changjicum* were the first and second branches that split off. Considering the distribution, Asia *Lycium* migrated eastward and resulted in the high diversification in East Asia (Fig. 4d). North American *Lycium* is genetically closest to *Lycium* species of the Asian clade (migrated from the New World[66]), and the migration from Asia into North America may have occurred via the Bering Strait (Fig. 4d). Alternatively, since both members of the eastern Asian clade and members of the north American clade share the same ancestor with species from the Middle East, the other possible route is migration westward to North America by long-distance dispersal (Fig. 4d). Nonetheless, our study suggests a potential migration event for *Lycium* into North America after establishment in the Old World first. Further studies including additional samples from eastern Europe and central Africa are probably needed to confirm this hypothetical route.

## Conclusion

Here, we have sequenced the genomes of 13 wolfberries (genus *Lycium*), many of which are common and important sources of food and medicine. The analysis of the *L. barbarum* and *L. ruthenicum* reference genomes and 30-fold coverage genome sequences of additional 11 Lycium species provides strong evidence of a previously suggested Solanaceae-specific whole-genome triplication, shared by all extant Solanaceae species. This WGT might be associated with the K/Pg boundary, as observed for several other plant lineages[67], and occurred shortly before the divergence of extant species of Solanaceae. Comparisons between *Lycium* and other Solanaceae showed that Solanaceae-specific genes are enriched in biological functions related to the biosynthesis of enzymes and secondary metabolites in response to various types of biotic and abiotic stresses. Furthermore, the expanded *FLC-like* MADS-box genes might correlate with the perennial flowering cycle of *Lycium*. Extremely low expression of *LC* and *CNR* orthologous genes during *Lycium* fruit development and ripening processes suggests functional diversification of these two genes between *Lycium* and tomato. Comparison of putative LBP biosynthetic genes in *L. barbarum* with other Solanaceae species suggests that SWEET genes are tandem duplicates, and mainly contribute to LBP biosynthesis in *Lycium*. Differential expression of genes involved in the lignin biosynthetic pathways between *Lycium* and tomato might illustrate woody and herbaceous differentiation in Solanaceae. Differential expression patterns of genes involved in anthocyanin and carotenoid biosynthetic pathways between *L. barbarum* with red fruits and *L. ruthenicum* with black fruits are also revealed.

Hybridization hotspots are enriched for genes that might be functioning in GSI pathways. We also provide evidence that *Lycium* in Asia, migrated from Africa, had a potential migration event from Asia into North America. Our results add functional insights into Solanaceae origins, evolution, and diversification.

## Methods

**DNA preparation and sequencing**. All of the diploid plant materials used in this study were collected from wild or farm, and transplanted to Goji Garden of National Wolfberry Engineering Research Center, Yinchuan, China (Supplementary Table 1). The haploid *L. barbarum* plant was developed by the pollen of diploid plant (Supplementary Note 1). Total genomic DNA was extracted with a modified cetyltrimethylammonium bromide method for Illumina and de novo sequencing and assembly. Five hundred bp paired-end libraries were constructed using the Illumina protocol. The genome size and heterozygosity were measured using GenomeScope[68] based on a 17 *K*-mer distribution. In addition, we constructed SMRT libraries using the PacBio 20-kb protocol (https://www.pacb.com/), and they were subsequently sequenced on the PacBio platform. The transcriptome of root, stem, leaf, flower, and fruit was sequenced on the Illumina platform.

**Genome assembly**. Canu[69] was used to correct errors in the original data. Flye v2.4.2 (ref. [70]) was used to assemble the corrected data. Because of the high error rate of de novo data, Indel, and SNP errors still existed in the assembly results. Thus, Arrow (https://github.com/PacificBiosciences/GenomicConsensus) was used to correct the assembly results. We compared the second-generation small fragment data with the assembly results and further corrected the assembly results with Pilon v1.22 (ref. [71]) to eliminate Indel and SNP errors. In addition, the assembly sequence was larger than the estimated genome size of *K*-mer, so we used the trimDup (Rabbit Genome Assembler: https://github.com/gigascience/rabbit-genome-assembler) to remove redundancy from the assembly results. To confirm the quality of the genome assembly, we performed a BUSCO v3 (https://anaconda.org/bioconda/busco)[15] assessment using single-copy orthologous genes.

**Resequencing and SNP calling**. DNA from single plant of 13 wolfberry accessions was extracted as described above. Paired-end libraries with insert sizes of 500 bp were constructed with the same protocol as de novo sequencing and sequenced on the HiSeq 4000 platform.

Reads were mapped with the BWA-MEM version 0.7.8 using the default parameters[72]. The obtained alignments were filtered using SAMtools version 0.1.19 (ref. [73]). Duplicated reads were marked using MarkDuplicated from Picard tools version 1.119 (http://broadinstitute.github.io/picard/). Finally, the SNPs for each genotype across the 13 accessions were identified using Genome Analysis Toolkit (GATK) version 2.4.9 (ref. [74]).

**Identification of repetitive sequences**. Tandem repeats across the genome were predicted using Tandem Repeats Finder (v4.07b, http://tandem.bu.edu/trf/trf.html). Transposable elements (TEs) were first identified using RepeatMasker (http://www.repeatmasker.org, v3.3.0), and RepeatProteinMask based on the Repbase TE library[75]. Thereafter, two de novo prediction software programs, RepeatModeler (http://repeatmasker.org/RepeatModeler.html) and LTR_FINDER[76], were used to identify TEs in the wolfberry genome. Finally, repeat sequences with identities ≥50% were grouped into the same classes.

**Gene prediction and annotation**. Three independent methods were used for gene predictions. Homologous sequence searching was performed by comparing protein sequences of six sequenced species against the wolfberry genome using the TBLASTN algorithm with parameters of *E*-value ≤ 1E − 5. Then, the corresponding homologous genome sequences were aligned against matching proteins using GeneWise v2.4.1 to extract accurate exon–intron information[77]. Three ab initio prediction software programs, Augustus v3.0.2 (ref. [78]), fgenesh[79], and GlimmerHMM[80], were employed for de novo gene predictions. Then, the homology-based and ab initio gene structures (Supplementary Fig. 29) were merged into a nonredundant gene model using GLEAN. Finally, the RNA-seq reads were mapped to the assembly using TopHat v2.0.11 (ref. [81]), and Cufflinks v2.2.1 (ref. [82]) was applied to combine mapping results for transcript structural predictions.

The protein sequences of the consensus gene set were aligned to various protein databases, including GO (The Gene Ontology Consortium), KEGG[83], InterPro[84], and Swiss-Prot and TrEMBL for the annotation of predicted genes. The rRNAs were identified by aligning the rRNA template sequences from the Rfam[85] database against the genome using the BLASTN algorithm at an *E*-value of 1E − 5. The tRNAs were predicted using tRNAscan-SE[86] and other ncRNAs were predicted by Infernal-0.81 software against the Rfam database.

**Genome evolution analysis**. Gene families present in the 19 genomes were identified using OrthoMCL[87]. Peptide sequences from 259 single-copy gene families were used to construct phylogenetic relationships and to estimate

divergence times. Alignments from MUSCLE were converted to coding sequences. Fourfold degenerate sites were concatenated and used to estimate the neutral substitution rate per year and the divergence time. PhyML[88] was used to construct the phylogenetic tree. The Bayesian Relaxed Molecular Clock approach was used to estimate the species' divergence times using the program MCMCTREE v4.0, which is part of the PAML package[89]. The "correlated molecular clock" and "JC69" models were used. Published tomato–potato (<20 and >10 Mya), and papaya–*Arabidopsis* (<90 and >54 Mya) were used to calibrate divergence times.

**Transcriptome sequencing and expression analysis.** Fruits from five different developmental stages during fruit ripening were collected independently for *L. barbarum* and *L. ruthenicum*. Total RNA was extracted according to the manufacturer's protocol. Illumina RNA-Seq libraries were prepared and sequenced on a HiSeq 2500 system following the manufacturer's instructions (Illumina, USA). Two biological replicates were performed for each sample. To estimate gene expression levels, clean reads of each sample were mapped onto the assembled genome to obtain read counts for each gene using TopHat and were normalized to FPKM reads[82].

**Distribution of substitutions per synonymous site ($K_s$).** $K_S$ age distributions for all paralogous genes (paranome) and paralogs located in collinear regions (anchor pairs) of the two *Lycium* genomes (Supplementary Fig. 7), four other Solanaceae genomes, and one Convolvulaceae genome (Supplementary Fig. 8) were calculated by the wgd software[90]. Orthologous $K_S$ distributions between *L. barbarum* and Solanaceae species—like *S. lycopersicum*, *N. tabacum*, and *P. axillaris*—and a Convolvulaceae species *I. nil*, and an outgroup species *C. canephora* were constructed by identifying one-to-one orthologs between species by selecting reciprocal BLASTP best hits[91], followed by $K_S$ estimation using maximum likelihood in the CODEML program of the PAML package (v4.4c)[88]. To identify collinear segments between the *L. barbarum* genome and the *V. vinifera* genome, i-ADHoRe (v3.0) was used with the parameters level_2_only=FALSE and anchor_points=3, enabling the ability to detect highly degenerated collinear segments resulting from more ancient large-scale duplications[92]. The $K_S$ values between anchor pairs calculated with the CODEML program, as above.

**Dating of the WGT event.** For each species in Solanaceae, we can calculate a rate of $K_s$ ($λ$) based on their divergence time with *Lycium* and the peak value in $K_s$ of differentiation, in brief, $λ = K_s$/time. The WGT time could be calculated by the WGT $K_s$ peak divided by $λ$. In the results, we estimated the time of Solanaceae WGT to be approximately 64.83–75.54 Mya (Supplementary Table 12).

**Identification of hybridization hotspots in the wolfberry genome.** The 20× Illumina reads were trimmed to remove the adaptors and low-quality bases using Trimmomatic[93] after quality control by FastQC[94]. The trimmed reads were mapped to the haploid reference genome using Bowtie2 (ref. [95]) with default parameters. The mapped reads were sorted, and duplicated reads were removed using SAMtools[73]. The Realigner Target Creator and Indel Realigner programs from the GATK package[74] were used for global realignment of reads around indels from the sorted BAM files. The HaplotypeCaller of the GATK was used to estimate the SNPs and Indels for putative diploids, using the default parameters. Low-quality variants were removed from the raw VCF file if they had DP < 2 or DP > 20, minQ < 20. SnpEff v3.6c (ref. [96]) was used to assign variant effects based on the gene model of the haploid genome. The genotype for each variant was extracted, and hybridization sites were counted using 200-kb sliding window along each chromosome of haploid genome. The density distribution (Supplementary Fig. 15) of the number of heterozygosity sites in each 200-kb sliding windows was plotted. According to the density distribution, the windows located at right tail of density map were treated as the hybridization hotspot regions, with the cut-off of >1000 heterozygous sites (0.5% heterozygosity rate) in each window. The hybridization hotspots were identified if the 200-kb sliding window had >1000 heterozygous sites. The genes located in hybridization hotspots were used as the gene set against the whole background gene model to perform the functional GO annotation and enrichment analysis in Blast2Go v4.1 (ref. [97]). The significance of enrichments was valued using Fisher's exact test.

**Homologous identification of SI-related gene families in Solanaceae.** Combining the morphological evidence that the pollen tube will stop growing before penetrating the ovary, we therefore infer the GSI in wolfberry. At present, several studies showed the key genes controlling the GSI mechanism are S-RNase/F-box (Solanaceae, Rosaceae, and Plantaginaceae) and PrsS/PrpS (Papaveraceae)[98]. Wolfberry belongs to the Solanaceae family, and the phenotype for pollen tube growth close to the S-RNase/F-box type seems to fit this scenario. The complete predicted proteome sequences of all the sequenced Solanaceae species were obtained from Solanaceae Genomics Network (solgenomics.net). The hidden Markov model (HMM) format file of Ribonuclease_T2(RNase-T2s, PF00445.18) downloaded from Pfam (pfam.xfam.org) was used to perform local searches in the proteome datasets by the HMM-based HMMER program 3.1b2 (ref. [99]). Sequences obtained were then aligned and manually adjusted in Multiple Alignment using Fast Fourier Transform (MAFFT)[100], using the E-INS-I alignment strategy for

sequence integrity analysis. Sequences with obvious errors were excluded from subsequent analyses.

**Gene tree construction, sequence characteristics, and isoelectric point in Solanaceae.** The amino acid sequences of Solanaceae RNase-T2s and S-RNase-related RNase-T2s from species with S-RNase-based SI[23,101,102] were used for phylogenetic analysis. Multiple sequence alignment was carried out using MAFFT with the E-INS-I strategy and was adjusted manually as necessary, and the phylogenetic tree was generated by the maximum likelihood method, using PhyML-SMS online version (http://www.atgc-montpellier.fr/phyml-sms/)[103]. The approximate likelihood-ratio test branch support, which was based on a Shimodaira-Hasegawa-like procedure, was estimated with a VT + G model.

There were three subfamilies in the RNase-T2s gene family, which were class I, class II and S-RNase. Genes of the S-RNase subfamily were the key factors of self-incompatibility, whose sequence characteristics lacked amino acid pattern 4 ([CG] P[QLRSTIK][DGIKNPSTVY][ADEIMNPSTV][DGKNQST])[22,104,105]. Depending on the amino acid pattern, MAFFT software was used for the sequence alignment, and sequences containing amino acid pattern 4 were obtained. All S-RNases have an isoelectric point (IP) between 8 and 10 (ref. [106]). The IPs of all peptides were calculated in ExPASy (https://web.expasy.org/protparam/).

**Sugar content determination.** Samples of 2.5 g fresh fruit were ground in liquid nitrogen. The powder was transferred to a flat-bottomed flask (volume 250 mL) containing 75 mL of 80% ethanol; the mortar was immediately washed with 25 mL of 80% ethanol twice and 2 mL of water once. The fluid was completely transferred to the flask, refluxed at 78 °C for 1 h, filtered while hot, and then the insoluble residue was washed with 25 mL of 80% ethanol three times. After filtration, the filtrate was brought up to 250 mL and used in the determination of free sugar. In addition, 150 mL of deionised water was added to the original flask, which was placed in a water bath for 1 h at 78 °C, and then the solution was vacuum-filtered through a filter paper while still hot. The filter paper and insoluble residue were washed four times with 20 mL of deionised water. The final solution volume was increased to 250 mL in a volumetric flask. Later, the polysaccharide and free sugar content were measured with a glucose standard by the sulfuric acid–phenol method.

**Polysaccharopeptide glycoprotein mass spectrometry.** As proteins are sensitive to high temperatures, glycoproteins were extracted at lower temperatures. Samples of 250 g of dried fruit were cryo-milled in liquid nitrogen. The powder was soaked in 1.0 L of 40 °C water for 24 h. The supernatant was recovered by centrifugation (10,000 r.p.m. for 10 min), and the precipitant was washed with 250 mL of water. The combined filtrate solution was concentrated by rotary evaporation to a volume of ~200 mL, and 1/5 volume of Sevag reagent (chloroform:butanol = 4:1) was used to remove free proteins seven to 12 times for 30 min each time. The supernatant was dialyzed (>7000 Dr) for 48 h at room temperature and was then concentrated by rotary evaporation. Polysaccharopeptides and AG proteins were precipitated using 2.5 times the volume of absolute ethanol overnight at room temperature. The precipitate was freeze-dried. Then, 0.1 g of precipitate was weighed and dissolved in 2 mL of water. A 10-μL sample was used for 12% sodium dodecyl sulfate–polyacrylamide gel electrophoresis. The specific electrophoretic bands were excised and used for mass spectrometry, and a protein database annotated the protein sequences.

**MADS-box and stress response related gene analysis.** MADS-box genes were identified by searching the InterProScan[107] results of all the predicted *L. barbarum* proteins. The predicted genes were manually inspected, and where gene predictions were short, MADS- or K-domains were only partially included. MADS-box domains consisting of 60 amino acids, identified by SMART[108] for all MADS-box genes, were then aligned using ClustalW. An unrooted neighbor-joining phylogenetic tree was constructed in MEGA 5 (ref. [109]) with default parameters. Bootstrap analysis was performed using 1000 iterations. The TPT, ZEP, and xanthoxin oxidase families were analyzed based on the same method. TBtools[110] was used to conduct the motif analysis.

**Biogeography analysis.** For high quality, the genome of haploid *L. barbarum* was used as the reference for alignment and SNP calling. The "mem" function in BWA 0.7.12 (ref. [71]) was used to conduct the alignment against the reference genome, and variant calling was carried out following GATK best practices, and the SAM file was reordered and sorted, and PCR duplications were removed by Picard. The realignment was conducted by GATK. For the SNP dataset, all callings supported by less than three reads or missing >30% of data were removed. Based on these filtering SNPs, the ML tree was reconstructed by IQtree with model TVM-R10. *C. annuum* was applied as an outgroup for its systematic position in Solanaceae and the PCA was conducted by GCTA. The input file for GCTA[111] was created by VCFTOOLS 0.1.14 (ref. [112]) and PLINK 1.9 (ref. [113]) based on filtering SNPs as the same dataset for phylogenetic reconstruction. Population size history has been reconstructed by PSMC[114]; the short insert size library from *L. barbarum* was aligned against the reference haploid genome by BWA and VCF was prepared

by a similar method as SNP calling. In PSMC, the evolutionary rate and the year per generation were set as 6.5e − 9 and 5, respectively.

**Statistics and reproducibility**. In RNA sequencing, three biological replicates of each tissue were evaluated. We collected three samples from fruit, root, stem, flower, and leaf of three individuals and the samples from the same tissue of were mixed as one replicate for RNA-seq.

**Reporting summary**. Further information on research design is available in the Nature Research Reporting Summary linked to this article.

## Data availability
Genome sequences and whole-genome assemblies have been submitted to the National Center for Biotechnology Information (NCBI) database under PRJNA640228.

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

## Acknowledgements

The authors acknowledge support from the Independent R & D project of Ningxia Academy of Agricultural and Forestry Sciences (NKYG14-15 and NXYG13-18), project of integrated development of science and technology innovation in primary, secondary and tertiary industries (YES-16-04), National Natural Science Foundation of China (31360360), Technical research plan project in Ningxia(11-12), the Project of Agricultural Breeding of New Wolfberry Varieties in Ningxia (2018NYYZ01); National Key R&D Program of China (Grant No.2019YFD1000400), The National Natural Science Foundation of China (no. 31870199), and the Key Laboratory of National Forestry and Grassland Administration for Orchid Conservation and Utilization Construction Funds (nos. 115/118990050;115/KJG18016A) for Z.-J.L. Y.V.d.P. acknowledges funding from the European Research Council (ERC) under the European Union's Horizon 2020 research and innovation program (grant agreement no. 833522) and from Ghent University (Methusalem funding, BOF.MET.2021.0005.01). Z.L. is funded by a postdoctoral fellowship from the research fund of UGent with number BOFPDO2018001701.

## Author contributions

Y.-L.C. managed the project. Y.-L.C and Z.-J.L. planned and coordinated the project; Y.-L.C., Z.-J.L., W.-C.T., C.-M.Y., Y.V.d.P., J.-Y.W., and D.-Y.Z. wrote the manuscript; Y.-L.L., Y.-F.F., W.A., J.H., G.-L.D., Y.-J.W., Z.-G.S., E.-N.J., P.-J.W., and K.Q. collected and grew the plant material; Y.-T.J., X.-D.L., B.L., X.Y., Y.H., W.-H.S., Q.L., J.-H.Z., Y.Y., B.Z., X.-Y.X., and X.-Y.L. prepared samples; Y.-Y.H., Y.-F.L., Y.-I.C., C.-K.L., W.-L.W., and H.-C.L. sequenced and processed the raw data; Y.V.d.P., Z.L., Z.-J.L., S.-R.L., and W.-C.T. analyzed the paleopolyploidy event; Y.-L.C., Y.-L.L., Z.-J.L., and W.-C.T. annotated the genome and analyzed gene families; Z.-J.L., W.-C.T., J.-Y.W., X.-K.M., Z.-W.W., X.Z., W.-Y.Z., M.-H.L., and S.-Q.Z. conducted genome evolution analysis; W.-C.T., Y.-F.L., Y.-Y.C., and C.-K.L. conducted the MADS-box gene analysis; W.-C.T., C.-M.Y., N.W., Y.-L.L., N.M., and K.Y. conducted transcriptome sequencing and analysis. Y.-L.L., K.Y., K.-C.T., N.M., T.K., T.I., and C.-M.Y. conducted polysarccharide analysis; J.-Y.W., W.-C.T., Z.-J.L., Y.-L.L., and Y.-L.C. conduct biogeography analysis. Y.-L.L., W.-C.T., and W.-H.S. conducted pigment biosynthesis pathways. X.-K.M., Z.-J.L., S.-C.N., and S.-C.Z. conducted self-incompatibility.

## Competing interests

The authors declare no competing interest.

## Additional information

[1]National Wolfberry Engineering Research Center, Ningxia Academy of Agriculture and Forestry Sciences, Yinchuan 750002, China. [2]Institute of Wolfberry Engineering Technology, Ningxia Academy of Agriculture and Forestry Sciences, Yinchuan 750002, China. [3]Department of Plant Biotechnology and Bioinformatics, Ghent University, 9052 Ghent, Belgium. [4]VIB Center for Plant Systems Biology, VIB, 9052 Ghent, Belgium. [5]Technology Center, Taisei Corporation, Yokohama, Kanagawa 245-0051, Japan. [6]Key Laboratory of National Forestry and Grassland Administration for Orchid Conservation and Utilization at College of Landscape Architecture, Fujian Agriculture and Forestry University, Fuzhou 350002, China. [7]Key Laboratory of Plant Resources Conservation and Sustainable Utilization, South China Botanical Garden, Chinese Academy of Sciences, Guangzhou 510650, China. [8]Faculty of Life and Environmental Sciences, University of Tsukuba, Tsukuba, Ibaraki 305-8572, Japan. [9]Bioproduction Research Institute, National Institute of Advanced Industrial Science and Technology (AIST), Tsukuba, Ibaraki 305-8562, Japan. [10]Graduate School of Science and Engineering, Saitama University, 255 Shimo-Okubo, Sakura-ku, Saitama 338-8570, Japan. [11]College of Life Sciences, Ritsumeikan University, Kusatsu, Japan. [12]Food and Fertilizer Technology Center for the Asian and Pacific Region, 14 Wenchow St., Taipei 10648, Taiwan, China. [13]College of Horticulture, Hebei Agricultural University, Baoding 071000, China. [14]Fujian Colleges and Universities Engineering Research Institute of Conservation and Utilization of Natural Bioresources, College of Forestry, Fujian Agriculture and Forestry University, Fuzhou 350002, China. [15]Orchid Research and Development Center, National Cheng Kung University, Tainan 701, Taiwan, China. [16]Institute of Tropical Plant Sciences and Microbiology, National Cheng Kung University, Tainan 701, Taiwan, China. [17]Department of Life Sciences, National Cheng Kung University, Tainan 701, Taiwan, China. [18]PubBio-Tech, Wuhan 430070, China. [19]Institute of Molecular Biology, National Chung Hsing University, Taichung 40227, Taiwan, China. [20]Centre for Microbial Ecology and Genomics, Department of Biochemistry, Genetics and Microbiology, University of Pretoria, Pretoria 0028, South Africa. [21]College of Horticulture, Nanjing Agricultural University, Nanjing, China. [22]These authors contributed equally: You-Long Cao, Yan-long Li, Yun-Fang Fan, Zhen Li, Kouki Yoshida. ✉email: youlongchk@163.com; chuanmingy@nchu.edu.tw; tsaiwc@mail.ncku.edu.tw; yves.vandepeer@psb.vib-ugent.be; zjliu@fafu.edu.cn

