## [Peer Review File · Communications Biology]

This manuscript has been previously reviewed at another Nature Research journal. This document only contains reviewer comments and rebuttal letters for versions considered at Communications Biology.

Reviewers' comments:

Reviewer #1 (Remarks to the Author):

This revised manuscript presents some improvements from the previous version. However, there are still quite a lot of issues that need to be addressed prior to publication.

Line 84-93: This part needs rewritten. First, much more than seven species have been sequenced so far. For each crop, multiple wild and cultivated species have been sequenced, e.g., wild tomatoes (*S. pennellii* (<https://www.nature.com/articles/ng.3046>) and *S. pimpinellifolium* (<https://www.nature.com/articles/s41467-020-19682-0>)), different pepper species (<https://genomebiology.biomedcentral.com/articles/10.1186/s13059-017-1341-9>)..... Second, it's not necessary to list the major findings from these genome sequencing papers.

Line 131-133: The assembled *L. ruthenicum* genome was not annotated and its quality was not evaluated. This must be done in order to make the genome more useful, especially as a reference genome (indicated in Line 557).

Line 138-139: I believe the BUSCO result here is for the predicted genes, not the genome (the BUSCO results for the genome were already described in Line 124-127).

Line 205-207: This sentence is poorly written. In addition, "RNase-T2s" was mentioned here without providing any background information.

Line 233: "0.5% heterozygosity rate" was used as the cutoff to identify hybridization hotspots. This seems not consistent with the estimated heterozygosity level of 1% (Line 120). This means that some of the identified hotspots had heterozygosity levels even lower than (only half of) the average heterozygosity level. Please explain. In addition, the y-axis of Extended Data Figure 4 is in the range of 0 to 25. What is the window size shown in this figure? Why only 25 heterozygous sites per window?

Line 246-256: These sentences are largely duplicated with the sentences in previous paragraphs. This can be substantially shortened, such as "Together the above results indicate that...".

Line 262-280: This paragraph is descriptive and the conclusion was not supported by the data. The expression data used to support the potential roles of the identified genes are from root, leaf, stem and flower (Fig. 3d; Please clearly explain the meaning of "red1", "R", "J"... in the legend), which can't support the potential roles of the genes in lateral root development.

Line 309-344: Same for this section, largely descriptive. The conclusion in Line 330-331 ("expended"->"expanded") is too speculative. The description in Line 337-344 seems to imply that these genes are involved in fruit development and ripening instead of flower development.

Line 360-361: The different expression patterns of these two genes from their tomato orthologs provide interesting information. However, both LC and CNR genes are members of large gene families, and it is necessary to make sure that the identified genes are exact homologs of tomato LC and CNR. Therefore, I highly suggest to perform both phylogenetic and synteny analyses to confirm their ortholog relationship.

I don't think Figure 4 is needed. The figure is mainly adopted and modified from previous published results, with listing several candidate genes identified in this study. Either remove this figure or convert it to a figure mainly presenting the results from this study.

Line 477-490: This paragraph was not clearly written. What data the authors were based on to infer that ATL and FACS catalyze the biosynthesis of ZD (Figure S16a)? Differential expression pattern can't

indicate their enzymatic functions.

Line 501-530: I found the conclusion made here were not supported by the data. For example, the statement "The Asian species showed significant divergence, while the Eurasian species clustered together and formed a single clade" (Line 503-504) is quite misleading. First, the tree (Extended Data Figure 8a) shows accessions from Asia (16 accessions), North America (4), Middle East (Asia) (1), and Europe /Africa / Asia (1), with "Asia" appearing in three different groups (why?). Second, I couldn't find the "Eurasia" group on the tree. Assuming it refers to "Europe /Africa / Asia", then there is only one accession (collected from Qinghai, China) from this group, and it doesn't make sense to state that "Eurasian species clustered together".

It's hard for me to understand how the authors can propose the migration scenario (Line 515-530) based on the phylogenetic tree and PCA.

I don't think the authors fully addressed my previous question "why accessions from the same species seem quite distantly related based on the branch lengths in the tree". In the current tree, the two *L. chinense* accessions and the two *L. dasystemum* accessions are quite distant on the tree. In addition, for all cultivars, the authors only provides the genus and the cultivar name, with no species name, e.g., *Lycium 'Mansheng'*. Why? In addition, I believe *Lycium 'Ningqi No.1'* is the reference *L. barbarum*. Please make this clear.

Line 531-553: Why these two paragraphs were put here? I couldn't see a clear connection of these two paragraphs with the above paragraph. I think they should fit better in the "Evolution of gene families" section.

There are still quite a few examples where the English should be improved. Here are several examples in Abstract:

- 1) Line 59, "Solanaceae family contains" -> "Solanaceae family, contains".
- 2) Line 62, "no woody representative has been available" -> "no genome sequences of woody representatives are available".
- 3) Line 64, "supports clear evidence of" -> "provides clear evidence supporting".
- 4) Line 65, "and dated shortly" -> "which occurred shortly"
- 5) Line 68, remove "in wolfberry".

Reviewer #2 (Remarks to the Author):

The authors have addressed many of the comments in the previous version of their manuscript but there are still a few issues that need to be addressed.

1. Many of the speculative claims linking gene copy number or expression patterns to biological traits have been removed or reduced, but there are still a few in the revised paper. For example, with regards to stress tolerance, the authors list out copy numbers of a few ABA genes and suggest differences may be related to stress tolerance in *Lycium* (lines 543-553). These findings are still not supported, and generally detract from the paper. More broadly, descriptive analyses of numerous genes for a dozen biological processes make it difficult to trudge through this paper. I understand that these analyses were probably time intensive, but they complicate an otherwise interesting paper, and would be better suited for future work.

Other sections that could be reduced or removed include: (1) water uptake and root system architecture (RSA), (2) ethylene biosynthesis in the fruit development and ripening section, and (3) Genes related to polysaccharide synthesis. This is simply a suggestion, but I think it would improve the paper.

2. The methods are reserved for the online version of the paper, but I see no reason why they are not included in the main text.

Response to Reviewers' comments:

Reviewer #1 (Remarks to the Author):

This revised manuscript presents some improvements from the previous version. However, there are still quite a lot of issues that need to be addressed prior to publication.

Comment 1

Line 84-93: This part needs rewritten. First, much more than seven species have been sequenced so far. For each crop, multiple wild and cultivated species have been sequenced, e.g., wild tomatoes (*S. pennellii* (<https://www.nature.com/articles/ng.3046>) and *S. pimpinellifolium* (<https://www.nature.com/articles/s41467-020-19682-0>)), different pepper species (<https://genomebiology.biomedcentral.com/articles/10.1186/s13059-017-1341-9>). Second, it's not necessary to list the major findings from these genome sequencing papers.

Response

In the revised version of our manuscript, we have rewritten this section considering the reviewers' comments and deleted the major findings from these genome sequencing papers (please see page 2, lines 85–87). The references of additionally sequenced species have been added:

4. Zhou, Q. *et al.* Haplotype-resolved genome analyses of a heterozygous diploid potato. *Nat. Genet.* **52**, 1018–1023 (2020).
6. Bolger, A. *et al.* The genome of the stress-tolerant wild tomato species *Solanum pennellii*. *Nat. Genet.* **46**, 1034–1038 (2014).
7. Wang, X. *et al.* Genome of *Solanum pimpinellifolium* provides insights into structural variants during tomato breeding. *Nat. Commun.* **11**, 5817 (2020).
9. Kim, S. *et al.* New reference genome sequences of hot pepper reveal the massive evolution of plant disease-resistance genes by retroduplication. *Genome Biol.* **18**, 210 (2017).

Comment 2

Line 131-133: The assembled *L. ruthenicum* genome was not annotated and its quality was not evaluated. This must be done in order to make the genome more useful, especially as a reference genome (indicated in Line 557).

Response

For the revised version of our manuscript, we have now annotated the *L. ruthenicum* genome and evaluated the annotation quality (please see page 3, lines 129–130, 133–135 and **Supplementary Tables 3, 6–8**).

Comment 3

Line 138-139: I believe the BUSCO result here is for the predicted genes, not the genome (the BUSCO results for the genome were already described in Line 124-127).

Response

Yes, we agree this was a little confusing. We have now added the BUSCO result for the assembled genome, and also changed the BUSCO result for the predicted genome (please see page 3,4, lines 133–136 and **Supplementary Table 7**).

Comment 4

Line 205-207: This sentence is poorly written. In addition, “RNase-T2s” was mentioned here without providing any background information.

Response

In the revised version of our manuscript, we have rephrased this sentence. Please see page 7, lines 222–224 for the revised sentence.

Comment 5

Line 233: “0.5% heterozygosity rate” was used as the cutoff to identify hybridization hotspots. This seems not consistent with the estimated heterozygosity level of 1% (Line 120). This means that some of the identified hotspots had heterozygosity levels even lower than (only half of) the average heterozygosity level. Please explain. In addition, the y-axis of Extended Data Figure 4 is in the range of 0 to 25. What is the window size shown in this figure? Why only 25 heterozygous sites per window?

Response

We first plotted the density distribution (see Fig S11 below) of the number of heterozygous sites in 200-kb sliding windows. Based on the density distribution pattern, we selected those windows at the right tail of the density map as the hybridization hotspot regions. This method let us choose windows with more than 1000 heterozygous sites (0.5% heterozygosity rate) in each window as the cutoff to identify the heterozygosity hotspots. We have modified the corresponding method part in the revised manuscript (page 22, lines 650–655).

The reported heterozygosity level of 1% is based on a Genome Survey analysis using resequenced Illumina data of the diploid genome. It was not consistent with our analysis using genotype-calling method. This is because our analysis of the heterozygous sites was identified using the genotype calling of a diploid individual using 20 X Illumina reads mapping to the haploid reference genome. By this method, our estimated average heterozygosity rate is 0.4% that should be lower than the estimation using the Genome Survey method for the whole diploid genome. But this will not influence the robustness of our results for comparing the difference of heterozygosity rate among different chromosomes using a sliding windows method.

The window is 200-kb as shown in Extended Data Figure 6. We are sorry the Extended Data Figure 6 used the wrong column of data for the y-axis. Previously, we used the density of heterozygous sites (Average number of heterozygous sites per 1kb) per each 200-kb window as the value of y-axis. So, it was ranging from 0 to 25. We have corrected the figure in the revised version, using the number of heterozygous sites per 200-kb window as the value of y-axis. Now the y-axis values range from 0 to 5000 per window.

Supplementary Figure 10. The density distribution of the number of heterozygosity sites in each 200-kb sliding window. The vertical line shows the cutoff line with 1000 heterozygous sites (0.5% heterozygosity rate).

Comment 6

Line 246-256: These sentences are largely duplicated with the sentences in previous paragraphs. This can be substantially shortened, such as “Together the above results indicate that...”.

Response

In the revised version of our manuscript, we have shortened this part according to the referees' suggestion. Please see the revised part as page 8, line 267.

Comment 7

Line 262-280: This paragraph is descriptive and the conclusion was not supported by the data. The expression data used to support the potential roles of the identified genes are from root, leaf, stem and flower (Fig. 3d; Please clearly explain the meaning of “red1”, “R”, “J”... in the legend), which can't support the potential roles of the genes in lateral root development.

Response

In the revised version of our manuscript, we have removed this section.

Comment 8

Line 309-344: Same for this section, largely descriptive. The conclusion in Line 330-331 (“expended”->”expanded”) is too speculative. The description in Line 337-344 seems to imply that these genes are involved in fruit development and ripening instead of flower development.

Response

In the revised version, “expended” has been changed to “expanded”. We also changed the tone to not over-interpret (Please see page 10, lines 319–322).

In the first paragraph of this section, we compared the number of MADS-box genes among *Lycium*, tomato and potato. In addition, we moved previous lines 337–344 to the next section for fitting the title of the section (Please see page 11, lines 343–351).

Comment 9

Line 360-361: The different expression patterns of these two genes from their tomato orthologs provide interesting information. However, both LC and CNR genes are members of large gene families, and it is necessary to make sure that the identified

genes are exact homologs of tomato LC and CNR. Therefore, I highly suggest to perform both phylogenetic and synteny analyses to confirm their ortholog relationship.

Response

Both the phylogenetic (**Supplementary Figure 14**) and synteny analyses (**Supplementary Figure 15**) showed that the *Lycium* CNR is orthologous to the tomato CNR. In addition, the *Lycium* LC also is orthologous to the tomato LC which was supported by both phylogenetic (**Supplementary Figure 16**) and synteny analyses (**Supplementary Figure 17**).

Supplementary Figure 14. Phylogenetic analysis of the *Lycium* LC gene and its homologous genes. The red rectangle denotes the LC gene from *Lycium*. The blue arrow denotes the LC gene from *Solanum lycopersicon*.

Supplementary Figure 15. Co-linear alignment of *Solanum lycopersicum* and *L. barbarum*. The red color of genes in the alignment denotes the tomato *LC* and putative *Lycium LC* genes. The grey links connect orthologues between partial regions of *S. lycopersicum* Chromosome 2 and *L. barbarum* Chromosome 12.

Supplementary Figure 16. Phylogenetic analysis of the *Lycium CNR* gene and its homologous genes. The red rectangle denotes the *CNR* gene from *Lycium*. The blue arrow denotes the *CNR* gene from *Solanum lycopersicon*.

Supplementary Figure 17. Co-linear alignment of *Solanum lycopersicum* and *L. barbarum*. The red color of genes in the alignment denotes the tomato *CNR* and putative *Lycium CNR*. The grey links connect orthologues between partial regions of *S. lycopersicum* Chromosome 2 and *L. barbarum* Chromosome 12.

Comment 10

I don't think Figure 4 is needed. The figure is mainly adopted and modified from previous published results, with listing several candidate genes identified in this study. Either remove this figure or convert it to a figure mainly presenting the results from this study.

Response

In the revised version of our manuscript, we have moved the original Figure 4 to Extended Data Figure 9. The original Figure 4 and the original Extended Data Figure 7b have been converted to the new Extended Data Figure 9. In addition, the original Extended Data Figure 9a, showing the potential candidate genes identified through the transcriptome, has been moved to the new Figure 3.

Comment 11

Line 477-490: This paragraph was not clearly written. What data the authors were based on to infer that ATL and FACS catalyze the biosynthesis of ZD (Figure S16a)? Differential expression pattern can't indicate their enzymatic functions.

Response

In the revised version of our manuscript, we have rewritten this part and deleted the related information on the ATL gene. **Supplementary Figure 19** and

Supplementary Table 28 have also been revised. Please see page 15, lines 454–469 for the revised text. Three references of FACS have been added as follows:

66. Jiang, Y., Chan, C. H. & Cronan, J. E. The soluble acyl-acyl carrier protein synthetase of *Vibrio harveyi* B392 is a member of the medium chain acyl-CoA synthetase family. *Biochemistry* **45**, 10008–19 (2006).
67. Weimar, J. D., DiRusso, C. C., Delio, R. & Black, P. N. Functional role of fatty acyl-coenzyme A synthetase in the transmembrane movement and activation of exogenous long-chain fatty acids. Amino acid residues within the ATP/AMP signature motif of *Escherichia coli* FadD are required for enzyme activity and fatty acid transport. *J. Biol. Chem.* **277**, 29369–76 (2002).
68. Shockey, J. M., Fulda, M. S. & Browse, J. A. Arabidopsis contains nine long-chain acyl-coenzyme a synthetase genes that participate in fatty acid and glycerolipid metabolism. *Plant Physiol.* **129**, 1710–22 (2002).

Comment 12

Line 501-530: I found the conclusion made here were not supported by the data. For example, the statement “The Asian species showed significant divergence, while the Eurasian species clustered together and formed a single clade” (Line 503-504) is quite misleading. First, the tree (Extended Data Figure 8a) shows accessions from Asia (16 accessions), North America (4), Middle East (Asia) (1), and Europe /Africa / Asia (1), with “Asia” appearing in three different groups (why?). Second, I couldn’t find the “Eurasia” group on the tree. Assuming it refers to “Europe /Africa / Asia”, then there is only one accession (collected from Qinghai, China) from this group, and it doesn’t make sense to state that “Eurasian species clustered together”.

It’s hard for me to understand how the authors can propose the migration scenario (Line 515-530) based on the phylogenetic tree and PCA.

Response

We agree with the reviewer that our initial writing could be somewhat misleading and have changed the text in the revised part of our manuscript. We clarified the relationship between the three Asian clades. The phrasing about the “Eurasia group” was also revised.

We also agree with the second remark and that the evidence from the phylogenetic tree and the PCA do not strongly support the scenario we envisioned. So, we have

also rewritten this part and only present now the most important and convincing results.

Comment 13

I don't think the authors fully addressed my previous question "why accessions from the same species seem quite distantly related based on the branch lengths in the tree". In the current tree, the two *L. chinense* accessions and the two *L. dasystemum* accessions are quite distant on the tree. In addition, for all cultivars, the authors only provided the genus and the cultivar name, with no species name, e.g., *Lycium* 'Mansheng'. Why? In addition, I believe *Lycium* 'Ningqi No.1' is the reference *L. barbarum*. Please make this clear.

Response

Actually, the breeding history of the cultivars we included in this study is not clear and they are just all called by the same species name, although many of them have undergone complicated hybridization. Indeed, this situation cause that pattern, in which samples of same species cannot form a single clade. Since discussing all these undetermined cultivars is not the purpose of this genome paper, we have decided to remove all undetermined cultivars, and to only focus on the original species sampled in the wild, in the revised version of our manuscript.

Comment 14

Line 531-553: Why these two paragraphs were put here? I couldn't see a clear connection of these two paragraphs with the above paragraph. I think they should fit better in the "Evolution of gene families" section.

Response

We agree with the reviewer, have shortened this part and moved it to the section "Evolution of gene families".

Comment 15

There are still quite a few examples where the English should be improved. Here are several examples in Abstract:

- 1) Line 59, "Solanaceae family contains" -> "Solanaceae family, contains".
- 2) Line 62, "no woody representative has been available" -> "no genome sequences of woody representatives are available".

- 3) Line 64, “supports clear evidence of” -> “provides clear evidence supporting”.
- 4) Line 65, “and dated shortly” -> “which occurred shortly”
- 5) Line 68, remove “in wolfberry”.

Response

We have carefully checked the revised version of our manuscript again and have tried to improve the English overall.

Reviewer #2 (Remarks to the Author):

The authors have addressed many of the comments in the previous version of their manuscript but there are still a few issues that need to be addressed.

Comment 1

Many of the speculative claims linking gene copy number or expression patterns to biological traits have been removed or reduced, but there are still a few in the revised paper. For example, with regards to stress tolerance, the authors list out copy numbers of a few ABA genes and suggest differences may be related to stress tolerance in *Lycium* (lines 543-553). These findings are still not supported, and generally detract from the paper. More broadly, descriptive analyses of numerous genes for a dozen biological processes make it difficult to trudge through this paper. I understand that these analyses were probably time intensive, but they complicate an otherwise interesting paper, and would be better suited for future work.

Response

In the revised version of our manuscript, we have shortened the section on ABA gene analysis and moved this part to the section “Evolution of gene families” as also suggested by Reviewer 1.

Comment 2

Other sections that could be reduced or removed include: (1) water uptake and root system architecture (RSA), (2) ethylene biosynthesis in the fruit development and ripening section, and (3) Genes related to polysaccharide synthesis. This is simply a suggestion, but I think it would improve the paper.

Response

We agree with the reviewer and, to increase readability, have now shortened/revise or removed previous sections on “water uptake and root system architecture” section, “ethylene biosynthesis” in fruit development and ripening section and “Genes related to polysaccharide synthesis” section including the related figures.

Comment 3

The methods are reserved for the online version of the paper, but I see no reason why they are not included in the main text.

Response

In the revised version of our manuscript, we have included methods in the main text.

REVIEWERS' COMMENTS:

Reviewer #1 (Remarks to the Author):

The authors have addressed all my concerns. Here are some suggested minor changes (mostly cosmetic):

Line 84: "11" -> "several".

Line 222: ", which belong to" to ". S-RNases belong to the".

Line 347, "was significant at fruit developmental stages": This is very confusing. What does this exactly mean?

Lane 350, "in fruit developmental stage 2 during ripening stage": The writing is also confusing.

Line 459-460: Change "NCBI CD-Search shows that the gene (Lba07g02085) contains a ttLC_FACS_AEE21_like domain. The enzyme family with genes containing ttLC_FACS_AEE21_like domains can activate" to "NCBI CD-Search showed that the gene (Lba07g02085) encoded a protein containing a ttLC_FACS_AEE21_like domain. The enzyme family with proteins containing ttLC_FACS_AEE21_like domains can activate".

Line 463: Change "16 carbon" to "16-carbon" and "18 carbon" to "18-carbon".
"to be well bind with": hard to understand. Maybe change to "to be able to bind".

Line 466-467: Change "we identified the expression level of the gene FACS, and the fragments per kilobase million (FPKM) value in *L. barbarum* was" to "we that the expression level of FACS in *L. barbarum* was".

Line 468-470: Change "Thus, we inferred that FACS (Lba07g02085) participated in the carotenoid biosynthesis process and plays a key role in catalysing the biosynthesis of ZD" to "Thus, we inferred that FACS (Lba07g02085) could play a key role in catalysing the biosynthesis of ZD."

Line 494: "is first" -> "is the first".

Line 496-498, "The African Lycium originated from the South American Lycium, and it seems that the ancestor of the Asian Lycium was African": Not easy to understand. How about changing this sentence to "The African Lycium could be derived from the South American Lycium, while it could be the ancestor of the Asian Lycium"?

Line 505: "on East Asia" -> "in East Asia".

Line 511: "No matter which hypothesis is most likely, our study" -> "Nonetheless, our study".

Line 518: Change "11 30-fold coverage genomes of additional Lycium species" to "30-fold coverage genome sequences of additional 11 Lycium species".

Line 1126: change first LBP to "*L. barbarum* polysaccharide (LBP)". Change all "LBP content" to "LBP contents".

Line 1127: "was distinguished to" -> "was divided into". Better to explain the five stages (e.g., days post pollination, as well as the corresponding fruit pictures...).

Line 1132: No "pink" color could be found in the figure.

Reviewer #2 (Remarks to the Author):

The authors have addressed my previous comments and concerns

Reviewer #1 (Remarks to the Author):

The authors have addressed all my concerns. Here are some suggested minor changes (mostly cosmetic):

Line 84: “11” -> “several”.

Thank you for your comments. We have changed ‘11’ to ‘several’.

Line 222: “, which belong to” to “. S-RNases belong to the”.

Thank you for your comments. We have changed “, which belong to” to “. S-RNases belong to the”.

Line 347, “was significant at fruit developmental stages”: This is very confusing. What does this exactly mean?

Thank you for your comments. We have changed the sentence to “was high at fruit developmental stages”.

Lane 350, “in fruit developmental stage 2 during ripening stage”: The writing is also confusing.

Thank you for your comments. We have changed the sentence to “is mainly expressed in fruit developmental stage 2”.

Line 459-460: Change “NCBI CD-Search shows that the gene (Lba07g02085) contains a ttLC_FACS_AEE21_like domain. The enzyme family with genes containing ttLC_FACS_AEE21_like domains can activate” to “NCBI CD-Search showed that the gene (Lba07g02085) encoded a protein containing a ttLC_FACS_AEE21_like domain. The enzyme family with proteins containing ttLC_FACS_AEE21_like domains can activate”.

Thank you for your comments. We have changed these sentences as suggested.

Line 463: Change “16 carbon” to “16-carbon” and “18 carbon” to “18-carbon”.

“to be well bind with”: hard to understand. Maybe change to “to be able to bind”.

Thank you for your comments. We have changed these words as suggested.

Line 466-467: Change “we identified the expression level of the gene FACS, and the fragments per kilobase million (FPKM) value in L. barbarum was” to “we that the expression level of FACS in L. barbarum was”.

Thank you for your comments. We have changed these sentences as suggested.

Line 468-470: Change “Thus, we inferred that FACS (Lba07g02085) participated in the carotenoid biosynthesis process and plays a key role in catalysing the biosynthesis of ZD” to “Thus, we inferred that FACS (Lba07g02085) could play a key role in catalysing the biosynthesis of ZD.”

Thank you for your comments. We have changed these sentences as suggested.

Line 494: “is first” -> “is the first”.

Thank you for your comments. We have made revisions as suggested.

Line 496-498, “The African Lycium originated from the South American Lycium, and it seems that the ancestor of the Asian Lycium was African”: Not easy to understand. How about changing this sentence to “The African Lycium could be derived from the South American Lycium, while it could be the ancestor of the Asian Lycium”?

Thank you for your comments. We have made revisions as suggested.

Line 505: “on East Asia” -> “in East Asia”.

Thank you for your comments. We have made revisions as suggested.

Line 511: “No matter which hypothesis is most likely, our study” -> “Nonetheless, our study”.

Thank you for your comments. We have made revisions as suggested.

Line 518: Change “11 30-fold coverage genomes of additional Lycium species” to “30-fold coverage genome sequences of additional 11 Lycium species”.

Thank you for your comments. We have made revisions as suggested.

Line 1126: change first LBP to “L. barbarum polysaccharide (LBP)”. Change all “LBP content” to “LBP contents”.

Thank you for your comments. We have made revisions as suggested.

Line 1127: “was distinguished to” -> “was divided into”. Better to explain the five stages (e.g., days post pollination, as well as the corresponding fruit pictures...).

Thank you for your comments. We have made revisions as S1: young stage, 9 days post anthesis; S2: green stage, 15 days post anthesis; S3: turning stage, 21 days post anthesis; S4: red stage, 28 days post anthesis; S5: ripe stage, 35 days post anthesis.

Line 1132: No “pink” color could be found in the figure.

Thank you for your comments. We have changed ‘pink’ to ‘light orange’.

Reviewer #2 (Remarks to the Author):

The authors have addressed my previous comments and concerns

Thank you for your comments and acknowledgement to our manuscript.